# Periodontitis and Other Risk Factors Related to Myocardial Infarction and Its Follow-Up

**DOI:** 10.3390/jcm11092618

**Published:** 2022-05-06

**Authors:** Tania Seoane, Beatriz Bullon, Patricia Fernandez-Riejos, Juan Carlos Garcia-Rubira, Nestor Garcia-Gonzalez, Pablo Villar-Calle, Jose Luis Quiles, Maurizio Battino, Pedro Bullon

**Affiliations:** 1Department of Cardiology, Lucus Augusi University Hospital, 27003 Lugo, Spain; tania.seoane.garcia@gmail.com; 2Department of Stomatology, Dental School, Universidad de Sevilla, 41009 Sevilla, Spain; beatrizbullon@hotmail.com; 3Department of Clinical Biochemistry, Virgen Macarena Hospital, 41009 Sevilla, Spain; patricia.fernandez.sspa@juntadeandalucia.es; 4Department of Cardiology, Virgen Macarena Hospital, 41009 Sevilla, Spain; jgarcia134@us.es (J.C.G.-R.); nestorgargon@gmail.com (N.G.-G.); 5Division of Cardiology, Weill Cornell Medicine, New York, NY 10065, USA; pablovc90@hotmail.es; 6Department of Physiology, Institute of Nutrition and Food Technology ‘José Mataix’, Center for Biomedical Research, Universidad de Granada, 18071 Armilla, Spain; jlquiles@ugr.es; 7Department of Clinical Sciences, Faculty of Medicine, Università Politecnica delle Marche, 60126 Ancona, Italy; m.a.battino@staff.univpm.it

**Keywords:** chronic periodontitis, myocardial infarction, risk factors, exercise, energy intake, mayor cardiovascular adverse events, left ventricular ejection fraction

## Abstract

The main issue in the prevention of myocardial infarction (MI) is to reduce risk factors. Periodontal disease is related to cardiovascular disease and both share risk factors. The purpose of this study is to investigate whether periodontitis can be considered a risk factor for MI and common risk factors in a case–control study and in a prospective follow-up study in patients with MI. The test group (MIG) was made up of 144 males who had MI in the previous 48 h. The control group (CG) was composed of 138 males without MI. Both groups were subdivided according to the presence or absence of stage III and IV of periodontitis. General data; Mediterranean diet and physical activity screening; periodontal data; and biochemical, microbiological and cardiological parameters were recorded. ANOVA, Mann–Whitney U and Kruskal–Wallis statistical tests and binary logistic regression analysis were applied. No differences in anthropometric variables were observed between the four groups. The average weekly exercise hours have a higher value in CG without periodontitis. The number of leukocytes was higher in MIG, the number of monocytes was higher in CG and the number of teeth was lower in MIG with periodontitis. Adherence to the Mediterranean diet was higher in CG. *Porphyromonas gingivalis* and *Tannerella forsythia* were higher in CG with periodontitis and in MIG with and without periodontitis. At follow-up, the left ventricular ejection fraction (LVEF) data were better in the non-periodontitis group: 15 patients had Mayor Cardiovascular Adverse Events (MACE), 13 of them had periodontitis and 2 did not show periodontitis. Periodontitis, exercise, diet and smoking are risk factors related to MI. MACE presented in the ‘MI follow-up’ shows periodontitis, weight, exercise hours and dyslipidemia as risk factors. LVEF follow-up values are preserved in patients without periodontitis. Our data suggest that periodontitis can be considered a risk factor for MI and MACE in the studied population.

## 1. Introduction

Cardiovascular disease (CVD) is the leading cause of death worldwide, and acute myocardial infarction (MI) is the most outstanding result [1]. Atherosclerosis is the main pathogenic feature behind these diseases and is considered an inflammatory process [2]. Inflammation is a defense reaction to the aggression of microorganisms such as bacteria, physical insults or chemical products. Periodontitis (PER) is an infectious inflammatory disease caused by an interaction between host responses and microbial dysbiosis that results in a loss of alveolar bone loss and can lead to tooth loss if left untreated [3]. Observational studies support an association between periodontitis and cardiovascular disease [4,5,6]. However, other studies have shown controversial results [7,8], which could impact public health policies if new risk factors for MI are identified. PER is associated with multiple bacterial species, mainly *Bacteroidetes*, and, more specifically, with the genera *Porphyromonas*, *Prevotella*, *Tannerella* and *Capnocytophaga*. These bacteria are involved in the development of atherosclerosis in animals [9] and are associated with symptomatic atherosclerosis [10] and its markers [11] in humans. We must emphasize the importance of the oral cavity in systemic diseases, as it is part of the digestive tract and the high level of interindividual contamination through mouth-to-mouth contact [12]. In the past few decades, incredible efforts have been made to improve our knowledge of the risk factors associated with MI. However, the pathogenic mechanisms involved in the development of atherosclerosis remain unclear. One of the most important methods to prevent MI and one of the most effective strategies to perform this is to control cardiovascular risk factors [13]. Some data show that there are differences related to sex/gender in coronary artery disease, and specific risk factors have been identified [14,15,16]. Exposure to risk factors is very common among patients who have developed CVD, but exposure to the same risk factors is also very prevalent among people who have not developed coronary heart disease [17]. 

Treatment of MI has high-level time-critical conditions; the earlier patients have access to treatment, the higher the chances of survival. After the acute phase, to prevent a new event, secondary prevention remains essential. The adequate management of risk factors has successfully improved prognosis after MI. However, despite this progress, there is still a need to find innovative methods to maximize secondary prevention and to find any risk factors that could precede them [18]. Our goals were (1) to identify risk factors associated with MI in men with and without periodontitis and (2) to investigate which risk factors influence the prognosis of the progression of MI.

## 2. Materials and Methods

### 2.1. Study Participants

The test group (MIG) consisted of male patients who had a first episode of MI in the previous 48 h who attended the University Hospital, Sevilla, from February 2017 to February 2019. During the same period, the control group (CG) was recruited from male patients attending at the School of Dentistry of the University of Seville without a history of MI. Both groups were asked to participate in the study and signed an informed consent form. The protocol was approved by the Ethics Committee of the same hospital (1588-N-20, 21 December 2016). Both groups were divided into two other groups according to the presence or absence of PER. PER were diagnosed according to the stage III and IV definition of the 2017 World Workshop on Classification of Periodontal and Peri-Implant Diseases and Condition [19]: clinical attachment level ≥ 5 mm at 2 ≥ sites in ≥ 2 non-adjacent teeth. Severe PER was defined as 30% tooth with probing depth ≥ 4 at 2 ≥ interproximal sites. Furthermore, the number of sites with a clinical attachment level ≥ 5 mm and the number of teeth were recorded. MIG was followed for at least two years. 

The sample size was obtained considering the combination of two dichotomous variables: cardiovascular (yes/no) and PER (yes/no) and with the means of data for the clinical attachment level (CAL) (0.01, 0.27 and 0.07) and the main quantitative variables of the groups (0.4). A minimum sample size of 43 patients was obtained when the values were established at 5% and the power at 90% (for factor A: CVD) and 99% (for factor B: PER) and for interaction.

### 2.2. Clinical Data

General data were recorded: age, height, weight, body mass index (BMI), abdominal waist circumference and smoking status. We consider a smoker as someone who smoked at least one cigarette per day and nonsmokers as those who do not currently smoke.

### 2.3. Dietary Assessment 

The Mediterranean Diet Adherence Screener (MDA) [20] was used. It consists of 12 questions on frequency of food consumption and 2 questions on food intake habits considered characteristic of the Spanish Mediterranean diet. Each question was scored 0 or 1. The final MAD score ranged from 0 to 14 and was classified as Very low (range), Low, High and Very high.

### 2.4. Physical Activity

Based on Minnesota Leisure Time Physical Activity [21], we asked for the hours spent on walking, any sport, household activity and other physical activity per week and obtained the average weekly exercise hours.

### 2.5. Oral Examination

A calibrated explorer (PB) recorded the depth of the probing pocket at six sites per tooth as the distance between the gingival margin and the bottom of the pocket and the level of clinical attachment as the distance between the cemento–enamel junction and the bottom of the pocket. Bleeding on probing was also recorded after 10 s [22] at four sites around each tooth. 

### 2.6. Biochemical Parameters 

In CG, venous blood samples were obtained after a fasting period and in MIG within 48 h after the acute episode in the routine cardiac control. The following biochemical parameters were obtained: fasting glycemia, urea, creatinine, uric acid, GPT, GGT, cholesterol, triglycerides, HDL, LDL, VLDL, total cholesterol/HDL index, albumin, total bilirubin, phosphorus, calcium, HbA1c, TSH, iron, cortisol and CRP for all patients and high sensitivity troponine-1 and alkaline phosphatase for cardiac patients using routine analytical methods.

### 2.7. Microbiological Sample

In each patient, the three deepest periodontal pockets were selected. After the removal of the supragingival plaque and isolation of the area, paper points were inserted into the gingival sulcus for 30 s. The samples were pooled, stored in Eppendorfs and immediately processed. DNA was extracted in 24 h using the High Pure PCR Template Preparation Kit (Roche, Mannheim, Germany) and divided into aliquots for the microIDent PCR test (Hain Lifescience, Nehren, Germany). DNA-STRIP technology was used to record the species of *Porphyromonas gingivalis* (Pg), *Tannerella forsythia* (Tf), *Treponema denticola* (Td), *Prevotella intermedia* (Pi) and *Aggregatibacter actinomycetemcomitans* (Aa), as previously described. For the determination of analytical sensitivity, a series of plasmid dilutions was used containing target sequences detectable with micro-IDent^®^plus11. The following detection limits were determined: Aa specie: 50–100 genome equivalents/PCR or 1000–5000 genome equivalents/paper point; all other species: 500–1000 genome equivalents/PCR or 10,000–50,000 genome equivalents/paper point. The results were classified into 5 groups according to bioburden: not detectable (<104, Aa < 103), within the limit (104, Aa 103), increased (<105, Aa < 104), high (<106, Aa < 105) and very high (≥106, Aa ≥ 105). 

### 2.8. Cardiological Parameters

At baseline, the following cardiovascular risk factors were recorded in MIG: Previous history of hypertension, diabetes mellitus, dyslipidemia, smoking and history of ischemic cardiopathy. The left ventricular ejection fraction (LVEF) was visually estimated by agreement between two observers and categorized into 4 groups: preserved > 50%, slightly reduced 40–49%, moderately reduced 31–40% and severe < 30%. Platelet antiaggregant therapy: Clopidogrel, Prasugrel and Ticagrelor. Type of acute coronary syndrome that differentiates ST elevation. Location of coronary obstruction. Number of coronary vessels affected with severe stenosis, dividing them into ≤two and >three affected vessels. The patients were followed until February 2020, and the following parameters were recorded: LVEF, major adverse cardiac events (MACE) and death. MACE is defined as a composite of nonfatal MI nonfatal stroke, repeated revascularization or admission due to a cardiovascular event [23].

### 2.9. Statistical Analysis

SPSS 25.0 software (IBM, Armonk, NY, USA) was used for statistical analyzes. Descriptive statistics included means, standard deviations, medians, quartiles and percentages. The normal distribution of the quantitative data was verified using the Kolmogorov–Smirnov test. The statistical ANOVA test for parametric distribution and the Mann–Whitney U and Kruskal–Wallis tests for the nonparametric distribution were used. The post hoc test utilized for the parametric distribution was Student’s *t* test with the previous Levine test on the homogeneity of variances, and for nonparametric distribution, the Mann–Whitney U test was used. Categorical variables were analyzed with a chi-square test to determine the groups that make a difference, and the standardized residuals were corrected according to the Haberman method. Values of *p* < 0.05 were considered statistically significant. Binary logistic regression analysis was used to estimate the association between different variables and the risk of MI and MACE at follow-up. We used logistic regression analysis to model the probability of an event occurring based on other factors. It includes the Backward LR method that considers all confounding variables. It starts from all variables that, in the two-by-two crosses, have shown a relationship with the variable to be estimated; the method is step-by-step exclusion, eliminating in each iteration the variable for which its relationship with the one to be estimated presents less significance until only the most relevant is left.

## 3. Results

Initially, 138 patients in the CG and 144 in the MIG were explored, with a later exclusion of 26 patients in CG and 13 patients in MIG due to missing data. The final sample size for CG was 112 patients (66 with PER and 46 without periodontitis (NOPER) and for MIG 131 patients (85 with PER and 46 NOPER) (Figure 1). During MIG follow-up, one patient died and four patients were lost.

### 3.1. General Variables

Table 1 shows the general characteristics of the four groups. The CG patients were older than MIG. No differences in anthropometric variables were observed. The average weekly exercise hours have a high value in CGNOPER compared to the other three groups and also for both CGs compared to MIG. The number of leukocytes was higher in MIG than in both CG, and the number of monocytes was low in CG compared to MIG. The number of teeth was lower in MIGPER. Smoking was high in MIG. MDA shows differences in low adherence and high adherence levels; thus, CG shows a diet closer to Mediterranean parameters than MIG.

The main parameters that define periodontitis show higher values in patients with periodontitis from CG and MIG. Moreover, severe PER presents the highest percentages in MIG (Table 2). 

### 3.2. Biochemical Variables

Differences between CG and MIG were observed for creatinine, GPT, total cholesterol, HDL, total cholesterol/HDL, albumin, total bilirubin, calcium, TSH and CRP. Cortisol shows the lowest figure in CGNOPER (Table 3). 

### 3.3. Microbiological Variables

The percentages of the different bacteria show statistical differences for Pg and Tf. Lower very high values (≥106, Aa ≥ 105) for Pg and Tf were observed in CGNOPER, the other three groups: CGPER and both MIG showed high values (Table 4).

### 3.4. Cardiac Data

The analysis of cardiac data in MIG does not show differences between NOPER and PER groups in cardiovascular risk factors (Table 5). Furthermore, when we analyzed cardiac variables of MI, no differences were observed. In follow-up, LVEF data were better in NOPER (Table 6). Fifteen patients had MACE (thirteen with PER and two without PER).

The results of the binary logistic regression (Table 7) show that the variables that show the probability related to MI are as follows: number of sites with CAL ≥ 5 mm (X1), severe PER (X2), average weekly exercise hours (X3), percentage of MDA (X4) and smoker (X5). These are risk factors related to an infarction with a sensitivity of 77.2% and a specificity of 76.1% (cutoff point of predicted probability = 0.5) and according to the following formula: *p* = 1/(1 + e^−(1 + e − (4.484 − 0.033 × X1 − 1.525 × X2 − 0.082 × X3 − 0.047 × X4 + 1.290 × X5)^).

The results of the binary logistic regression (Table 8) show that the variables that can be considered as risk factors for MACE are the following. PER (X1), weight (X2), average weekly exercise hours (X3) and dyslipidemia (X4). These are MACE-related risk factors with a sensitivity of 72.7% and a specificity of 85.6% (cutoff point of the predicted probability = 0.2) and according to the following formula: *p* = 1/(1 + e^−(3.966 − 3.432 × X1 + −0.091 × X2 + 0.212 × X3 + 1.136 × X4)^).

## 4. Discussion

Our results show that periodontitis is related to MI, as demonstrated in the literature [24,25,26]. Furthermore, another main finding of this study is that periodontitis should be considered as a risk factor in MI follow-up, such as Eppendorf. In MIG, we found no differences in cardiovascular risk factors between periodontitis patients and no periodontitis patients including hypertension, diabetes mellitus, dyslipidemia and smoking. These risk factors are the most prevalent in some published studies [27,28]. In addition, other risk factors such as waist circumference and HbA1c are not different between the groups. Therefore, our data highlight that patients with those same systemic conditions could have another risk factor, such as PER. Some authors show that CVD patients do not have any of the classic risk factors and 60% of them have not experienced MI [29]. We agree with some authors who propose an inflammatory genetic component in coronary artery disease [30]; our data suggest that one of the infectious diseases that can stimulate the inflammatory statement is periodontitis through oral bacteria. 

The percentages of cardiac variables are not statistically different between MIGNOPER and MIGPER. In the MIG follow-up, the only variable that had statistical differences between MIGPER and MIGNOPER is LVEF. LVEF is one of the most important parameters for verifying the functional statement of cardiac function, so it is very important for the prognosis of patients with MI to consider all factors that could influence it. The number of MACE and two types of MACE, non-fatal MI and admission due to CV events, was high in MIGPER but not statistically different. These data reveal a worse recovery after the acute event, which could be explained by more severe periodontal conditions. We must note that MIGs were followed for at least one year and patients, due to cardiac conditions, were not treated for PER. We can conclude that PER can alter cardiac recovery and influences left ventricular remodeling and future cardiac events. 

When we analyzed the results of binary logistic regression, a strong association was observed between average weekly exercise hours, MDA and smoking in MI patients. The binary logistic regression analysis shows an association between PER, weight, average weekly exercise hours, dyslipidemia and MACE. Therefore, the number of teeth and PER, which indicate unhealthy periodontal conditions, are part of factors that can be related to MI and affect cardiac evolution. Missing teeth is associated with an increased risk of MI and discriminates patients with high cardiovascular risk [31]; low physical activity with an inadequate diet that produces a metabolic syndrome is associated with a high risk of major adverse cardiovascular events [32]. Furthermore, exercise-based cardiac rehabilitation is essential in cardiac patients’ recovery of cardiac patients [33]; a recent cohort study tried to associate the impact of periodontitis and subgingival bacteria on recurrent CVD events such as MI, stroke and death caused by CVD. The results cannot show an association between the periodontal bacteria that we have studied, although an association between periodontitis and more CVD events is shown. [34].

The relationship between MI and PER has been extensively studied. A recent meta-analysis of cohort studies indicates that PD is modestly associated with the risk of MI, especially in women [24]. In a 13-year prospective study, severe PER was shown to be associated with an increased incidence of coronary heart disease regardless of the established cardiovascular risk factors [35]. A review reports on the proceedings of the workshop jointly organized by the EFP and the World Heart Federation (WHF). They concluded that there is epidemiological evidence for significant associations between periodontitis and CVD and recommended that dentists, physicians and patients visit dental and medical practices [36].

The general characteristics of the patients show that the most important factor that can distinguish the different conditions is physical activity and a healthy diet and not the anthropometric variables. We have to consider that we usually have a hypercaloric diet in our countries that results in a specific weight; our results show that the more important factors are the dietary components and their consumption. Various diet indices have shown that unhealthy eating patterns are associated with severe PER [37,38] and others are associated only with obese men [39]. Mediterranean components such as salads, fruits and vegetables were associated with a lower level of clinical attachment level [40], and MDA produces a significant decrease in the saliva levels of *Pg*, *Pi* and *Tp* [41]. Physical activity has been associated with PER in general populations [42,43] and can improve periodontal health in patients with T2 diabetes mellitus [44] and obese women [45]. The Mediterranean diet and moderate physical activity are some of the most important recommendations for the primary prevention of CVD [46]. Smoking has been considered a common risk factor for PER and cardiovascular disease [47] and we show that the proportion of current smokers was high in MIG. The number of teeth has the lowest value in MIGPER, indicating that both conditions have an increasing effect, this result is in agreement with previous data that can discriminate patients with a high risk of MI [31]. 

We show that the number of leukocytes was high in both MIGs, as previously reported [48]. An association of leukocyte counts and systemic inflammation was demonstrated in patients with PD [49]. Our data show that the number of monocytes is low in both MIGs. This could be explained by the margination of monocytes to the coronary vasculature shown in patients after coronary artery reperfusion, which could potentially contribute to the drop in circulating monocytes [50].

Some of the biochemical parameters show differences between MIG and CG as consequences of MI conditions, but they cannot differentiate if they suffer from PER or NOPER. CRP is the only variable that increases in MIGPER that demonstrates the possible summation effects of cardiac and periodontal conditions. CRP is synthesized by the liver in response to any type of infection and is mainly stimulated by interleukin-6. It is a short pentatrexin, a multimeric pattern recognition protein that is considered a relatively moderate predictor of coronary heart disease [51], but CRP is also a strong predictor of cardiovascular mortality during the year after an MI event [52]. As we discuss below, our patients with PER suffered more MACE and worse LVEF at follow-up.

Our microbiological results show lower values for *Pg*, *Tf*, *Td* and *Pi* in CGNOPER and similar high figures in both MIG and CGPER. Therefore, cardiac patients have gingival microbiota compatible with periodontopathogen bacteria. *Pg* and *Tf*, the main periodontopathogen bacteria related to PER, are present in large numbers in MIGNOPER, so they are not related to periodontal conditions. We can deduce some deleterious effects on cardiovascular conditions. This aspect agrees with previous data. The presence of all these bacteria was detected in subgingival samples of cardiovascular patients who required endarterectomy [53,54]. One of them, *Pg*, has been associated with the development of atherosclerosis in animal models [55]. The oral microbiota is part of the digestive tract and is related to the gut microbiota. Research in animals shows that oral *Pg* infection disrupts oral and cecal microbiomes, altering their community membership and functional potential [56]. 

Our study has some limitations. Our patients are Spanish; share specific diet, environmental and cultural characteristics that are different from other populations; and the results should be evaluated based on these issues. We studied only male patients and the results cannot apply to women. The patients with MIG were explored 48 h after suffering from an MI, so they are patients who survived and had a cardiac recovery; we cannot explore those who did not survive and perhaps they had the worst cardiac conditions.

## 5. Conclusions

Periodontitis, exercise, diet and smoking are risk factors related to MI. Mayor Cardiovascular Adverse Events (MACEs) presented in ‘MI follow-up’ show periodontitis, weight, exercise hours and dyslipidemia as risk factors. LVEF follow-up values are preserved in patients without periodontitis. Periodontitis should be considered an important risk factor in MI follow-up.

## Figures and Tables

**Figure 1 jcm-11-02618-f001:**
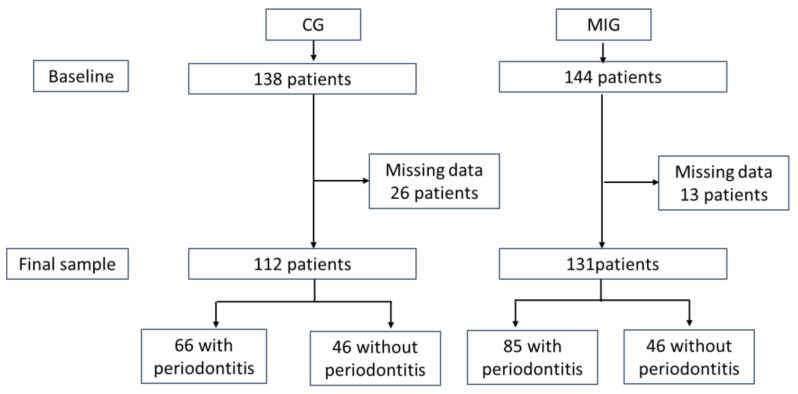
Study flow diagram.

**Table 1 jcm-11-02618-t001:** General characteristics of the four groups, means (standard deviation), medians, quartiles and percentages. Chi-square, ANOVA, Mann–Whitney U and Kruskal–Wallis test.

Variables		CG (112)		MIG (131)	
		NOPER ^a^ (46)	PER ^b^ (66)	NOPER ^c^ (46)	PER ^d^ (85)
Age		61.85 (8.85) ^c,d^	61.74 (8.05) ^c,d^	56.04 (9.58)	58.48 (7.41)
Height		172.80 (7.38)	171.44 (6.24)	172.65 (6.40)	172.55 (5.76)
BMI		27.95 (3.45)	27.37 (4.39)	29.17 (4.29)	28.00 (4.08)
Abdominal circumference		106.57 (9.89)	106.11 (11.61)	107.39 (8.51)	108.07 (10.82)
Hemoglobin		14.40 (1.97)	15.00 (1.58)	15.18 (1.32)	15.14 (1.76)
Monocytes		8.53 (2.95) ^c,d^	8.69 (2.61) ^c,d^	5.50(3.02)	5.87 (3.27)
Weight	Q1	75.00	72.00	80.00	74.00
	Q2 (Median)	81.50	79.50	85.00	82.00
	Q3	89.00	87.25	93.50	90.00
Average weekly exercise hours	Q1	4.38	2.00	0.00	0.00
	Q2 (Median)	7.75 ^b,c,d^	6.00 ^a,c,d^	3.15 ^a,b^	2.00 ^a,b^
	Q3	14.00	10.63	8.00	7.00
Leukocytes	Q1	5.57	5.79	8.63	9.42
	Q2 (Median)	6.43 ^c,d^	6.97 ^c,d^	9.97 ^a,b^	10.88 ^a,b^
	Q3	7.56	8.71	12.53	14.11
Number of teeth	Q1	20.00	16.00	18.75	9.00
	Q2 (Median)	24.00 ^b,d^	21.00 ^a,d^	22.00 ^d^	18.00 ^a,b,c^
	Q3	26.00	25.00	26.00	22.50
Smoking	% Current	15.2 *	27.3 *	59.1 *	57.1 *
	% None	84.8 *	72.7 *	40.9 *	42.9 *
MDA	% Very low	0.0	0.0	6.5	5.9
	% Low	24.4 *	27.3 *	56.5 *	56.5 *
	% High	48.9	50.0	30.5	32.9
	% Very high	26.7 *	22.7 *	6.5	4.7 *

BMI: body mass index; MDA: Mediterranean diet adherence. The statistical differences of the post hoc test between the groups were expressed as ^a, b, c, d^, representing each of the 4 groups. * Statistical significance *p* < 0.05. Age, height, BMI, abdominal circumference, hemoglobin and monocytes and ANOVA test. Weight, average weekly exercise hours, leukocytes and number of teeth, Kruskal–Wallis and Mann–Whitney test. Smoking and MDA and Chi-square test.

**Table 2 jcm-11-02618-t002:** Periodontal parameters of the four groups, Chi-square, Kruskal–Wallis and Mann–Whitney U test.

Variables		CG (112)		MIG (131)	
		NOPER ^a^ (46)	PER ^b^ (66)	NOPER ^c^ (46)	PER ^d^ (85)
PD	Q1	2.38	2.56	2.33	2.49
	Q2 (Median)	2.49 ^b,d^	2.79 ^a,c^	2.56 ^b,d^	2.71 ^a,c^
	Q3	2.65	3.16	2.69	3.10
CAL	Q1	2.50	3.22	2.36	3.46
	Q2 (Median)	2.70 ^b,d^	3.75 ^a,c^	2.73 ^b,d^	4.22 ^a,c^
	Q3	2.87	4.97	2.90	4.90
BOP	Q1	0.00	1.74	0.00	1.23
	Q2 (Median)	1.93 ^b,d^	8.75 ^a,c^	3.15 ^b,d^	5.56 ^a,c^
	Q3	6.45	19.54	8.04	16.67
Severe PER	% No		45.5 *		24.7 *
	% Yes		54.5 *		75.3 *

PD: probing pocket depth; CAL: clinical attachment level; BOP: bleeding on probing. The statistical differences of the post hoc test between the groups were expressed as ^a, b, c, d^, representing each of the 4 groups. * Statistical significance *p* < 0.05.

**Table 3 jcm-11-02618-t003:** Biochemical parameters of the four groups, means and standard deviation, medians, quartiles, ANOVA and Kruskal–Wallis test.

Variables		CG		MIG	
		NOPER ^a^	PER ^b^	NOPER ^c^	PER ^d^
Urea		34.33 (7.28)	32.78 (7.65)	35.14 (9.58)	33.32 (10.94)
Uric acid		5.68 (1.10)	5.78 (1.53)	6.11 (1.64)	6.14 (1.45)
Total cholesterol		191.33 (41.65) ^c^	192.94 (41.02)	166.36 (46.31) ^a^	176.44 (51.55)
LDL		117.13 (35.61)	110.83 (37.88)	96.86 (35.67)	103.87 (40.46)
Total Cholesterol/HDL		4.15 (1.24) ^d^	4.04 (1.39) ^d^	4.49 (1.63)	4.92 (1.84) ^a,b^
Calcium		9.25 (0.42) ^c,d^	9.33 (0.43) ^c,d^	9.05 (0.53) ^a,b^	8.91 (0.74) ^a,b^
Iron		93.80 (32.59)	82.59 (29.91)	74.42 (33.51)	88.47 (54.66)
Cortisol		98.25 (60.45) ^b,c,d^	137.98 (53.50) ^a^	161.92 (77.47) ^a^	166.73 (102.93) ^a^
Alkaline phosphatase				76.47 (21.93)	67.94 (21.63)
Fasting glycemia	Q1	89.50	86.75	85.50	91.00
	Q2 (Median)	96.00	95.50	97.00	100.00
	Q3	107.50	113.25	113.00	119.50
Creatinine	Q1	0.78	0.72	0.80	0.78
	Q2 (Median)	0.88 ^b^	0.82 ^a,c,d^	0.90 ^b^	0.85 ^b^
	Q3	0.99	0.91	1.02	1.00
GPT	Q1	16.00	14.75	22.00	19.50
	Q2 (Median)	20.50 ^c,d^	20.50 ^c,d^	31.00 ^a,b^	32.00 ^a,b^
	Q3	29.00	30.50	46.00	50.50
GGT	Q1	20.75	22.00	21.00	19.75
	Q2 (Median)	23.00	34.00	38.00	33.50
	Q3	56.25	50.50	56.00	46.50
Triglycerides	Q1	78.25	93.00	87.50	96.00
	Q2 (Median)	110.00	112.00	128.00	139.00
	Q3	162.75	173.50	176.00	212.00
HDL	Q1	41.75	40.00	32.00	31.00
	Q2 (Median)	47.00 ^c,d^	47.00 ^c,d^	38.50 ^a,b^	36.50 ^a,b^
	Q3	57.00	60.25	44.00	44.00
VLDL	Q1	15.50	19.00	17.25	19.00
	Q2 (Median)	22.00	23.00	25.50	27.00
	Q3	31.50	35.00	35.00	42.00
Albumin	Q1	4.00	4.20	4.03	3.80
	Q2 (Median)	4.30 ^d^	4.50 ^c, d^	4.30 ^b^	4.10 ^a, b^
	Q3	4.50	4.60	4.60	4.60
Total bilirubin	Q1	0.40	0.40	0.41	0.40
	Q2 (Median)	0.50 ^c,d^	0.50	0.70 ^a^	0.60 ^a^
	Q3	0.63	0.60	0.80	0.90
Phosphorus	Q1	2.60	2.70	2.85	2.70
	Q2 (Median)	2.95	3.00	3.00	3.00
	Q3	3.43	3.30	3.55	3.40
HbA1c	Q1	5.45	5.40	5.48	5.50
	Q2 (Median)	5.60	5.70	5.70	5.80
	Q3	5.95	6.20	6.10	6.20
TSH	Q1	1.33	1.22	1.14	0.86
	Q2 (Median)	2.01 ^d^	1.70 ^d^	1.98 ^d^	1.23 ^a,b,c^
	Q3	2.97	2.83	2.57	1.84
CRP	Q1	1.41	0.87	0.71	1.37
	Q2 (Median)	2.39 ^d^	1.80 ^d^	1.68 ^d^	3.84 ^a,b,c^
	Q3	3.65	4.11	4.16	7.53
Troponin	Q1			15.00	24.85
	Q2 (Median)			40.00	65.50
	Q3			141.00	203.00

CRP: C-reactive protein; HcA1c: glycated hemoglobin; GPT: glutamate pyruvate transaminase; GGT: gamma-glutamyl transpeptidase; HDL: high-density lipoprotein; LDL: low-density lipoprotein; TC/HDL: total/high-density lipoprotein (HDL) cholesterol ratio; TSH: thyroid stimulating hormone; VLDL: very low-density lipoprotein; HbA1c: glycated hemoglobin. The statistical differences of the post hoc test between the groups were expressed as ^a, b, c, d^, representing each of the 4 groups. Urea, uric acid, total cholesterol, LDL, total cholesterol/HDL, calcium, iron, cortisol, alkaline phosphatase and ANOVA test. Fasting glycemia, creatinine, GPT, GGT, triglycerides, HDL, VLDL, albumin, total bilirubin, phosphorus, HbA1c, TSH, CRP, troponin, Kruskal–Wallis and Mann–Whitney U test.

**Table 4 jcm-11-02618-t004:** Microbiological parameters of the four groups expressed as bioburden categories and percentages; Chi-square test.

Variables	Categories	CG		MIG	
		NOPER	PER	NOPER	PER
*Aggregatibacter actinomycetemcomitans*	Not detectable	87.0	97.0	95.6	96.4
	Within the limit	6.5	0.0	0.0	1.2
	Increased	0.0	1.5	0.0	0.0
	High	2.2	0.0	0.0	0.0
	Very high	4.3	1.5	4.4	2.4
*Phorphyromona gingivalis*	Not detectable	50.1	31.8	46.7	33.3
	Within the limit	13.0 *	3.0	2.2	3.6
	Increased	13.0 *	3.0	6.7	4.8
	High	2.2	4.5	4.4	7.1
	Very high	21.7 *	57.7 *	40.0	51.2
*Tannerella forsythia*	Not detectable	34.7 *	10.6	13.3	16.7
	Within the limit	8.7	1.5	4.4	6.0
	Increased	19.6 *	7.6	13.3	4.8 *
	High	8.7	16.7	13.3	13.1
	Very high	28.3 *	63.6	55.7	59.4
*Treponema denticola*	Not detectable	34.7	18.2	15.6	19.0
	Within the limit	8.7	1.5	8.9	4.8
	Increased	17.4	9.1	11.1	13.1
	High	19.6	45.4	40.0	35.7
	Very high	19.6	25.8	24.4	27.4
*Prevotella intermedia*	Not detectable	60.9	56.0	51.2	58.3
	Within the limit	15.2	4.5	13.3	10.7
	Increased	17.4	15.2	8.9	10.7
	High	4.3	16.7	13.3	15.5
	Very high	2.2	7.6	13.3	4.8

Not detectable (<10^4^, Aa < 10^3^); Within the limit (10^4^, Aa 10^3^); Increased (<10^5^, Aa < 10^4^); High (<10^6^, Aa < 10^5^); Very high (≥10^6^, Aa ≥ 10^5^). * Statistical significance *p* < 0.05.

**Table 5 jcm-11-02618-t005:** Percentages of cardiovascular risk factors in MIG; Chi-square test.

Variables	Categories	MIG	
		NOPER	PER
Hypertension	Yes	43.5	48.2
	No	56.5	51.8
Diabetes Mellitus	Yes	17.4	23.5
	No	82.6	76.5
Dyslipidemia	Yes	43.5	40.0
	No	56.5	60.0
Smoking	Current	45.7	60.0
	None	32.6	16.5
	Former	21.7	23.5
Ischemic heart disease	Yes	8.7	11.8
	No	91.3	88.2

**Table 6 jcm-11-02618-t006:** Percentages of cardiac variables in MIG; Chi-square test.

Variables	Categories	MIG	
		NOPER	PER
LVEF on admission	Preserved	74.0	64.6
	Slightly reduced	17.4	22.4
	Moderately reduced	4.3	11.8
	Severely reduced	4.3	1.2
Antiplatelet therapy	No	8.7	2.4
	Clopidogrel 300 mg	13.0	11.8
	Clopidogrel 600 mg	13.0	8.2
	Clopidogrel 75 mg	0.0	2.4
	Prasugrel 60 mg	15.2	8.2
	Ticagrelor 180 mg	50.1	67.0
ACS	STEMI	84.8	75.3
	NSTEMI	15.2	24.7
Localization	Anterior	23.9	31.8
	Inferior	50.0	41.1
	Lateral	2.2	2.4
	Inferolateral	21.7	10.6
	Indeterminate	2.2	14.1
Number of vessels affected	≤Two	87.0	90.6
	>Three	13.0	9.4
LVEF in follow-up	Preserved	97.5 *	75.9 *
	Slightly reduced	0.0 *	20.7 *
	Moderately reduced	2.5	3.4
	Severely reduced	0.0	0.0
MACE	Yes	4.3	15.9
	No	95.7	84.1
Type of MACE	Non-fatal myocardial infarction	0.0	1.4
	Non-fatal stroke	50.0	0.0
	Repeated revascularization	50.0	46.1
	Admission due to CV event	0.0	38.5
EXITUS	Yes	0.0	1.2
	No	100.0	98.8

ACS: acute coronary syndrome; NSTEMI: non-ST elevation acute coronary syndrome; LVEF: eft ventricular ejection fraction; STEMI: ST segment elevation myocardial infarction; MACE: Mayor Cardiovascular Adverse Events. * Statistical significance *p* < 0.05.

**Table 7 jcm-11-02618-t007:** Results of the binary logistic regression analysis to estimate the association between the different variables and those who suffer from MI.

Variables	B	Standard Error	*p*-Value	OR	95% CI
Number of sites with CAL ≥ 5 mm	−0.033 *	0.011	0.002	0.968	0.948–0.988
No Severe PER	−1.525	0.600	0.011 *	0.218	0.067–0.706
Average weekly exercise hours	−0.082	0.030 *	0.006	0.921	0.869–0.976
Percentage MDA	−0.047 *	0.011 *	<0.001 *	0.954	0.934–0.975
Smoker	1.290	0.340	<0.001 *	3.631	1.865–7.069
Constant	4.484	0.930	<0.001 *	88.629	

MDA: Mediterranean diet adherence; B values of the exponent of the formula. * Statistical significance *p* < 0.05.

**Table 8 jcm-11-02618-t008:** Results of the binary logistic regression analysis to estimate the association between different variables and those who suffer from MACE.

Variables	B	Standard Error	*p*-Value	OR	95% CI
Periodontitis	−3.432	1.649	0.037 *	0.032	0.001–0.818
Weight	−0.091	0.047	0.052	0.913	0.999–1.2
Average weekly exercise hours	0.212	0.075	0.005 *	1.236	0.699–0937
Dyslipidemia	1.136	0.788	0.149	3.113	0.665–14.577
Constant	3.966	3.445	0.250	52.771	

B values of the exponent of the formula. * Statistical significance *p* < 0.05.

## Data Availability

The data presented in this study are available upon request from the corresponding author. The data are not publicly available due to Spanish privacy legislation.

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
