# Peer review of "Periodontitis and Other Risk Factors Related to Myocardial Infarction and Its Follow-Up"

_jcm, 2022, doi:10.3390/jcm11092618_

Round 1

Reviewer 1 Report

Although your paper is interesting, major improvements are necessary before publication. I suggest authors to consult STROBE guidelines for reporting the study. I also recommend submission to a native speaker for grammar and style review. 

Based on STROBE guidelines, please consider the following concerns:

  1. Indicate the study’s design with a commonly used term in the title or the abstract. This is a retrospective case-control study followed by a prospective follow-up of the participants included. Please consult a statistics for further clarification.
  2. In the abstract, you have stated that periodontal diseases  is a risk factor for cardiovascular diseases. If that is so, why are you performing this investigation?
  3. At the abstract, please clarify the goals of this study. I could understand that this study has two major goals: (1) to investigate if periodontal disease is a risk factor for myocardial infarction in a case-control study; (2) to investigate risk factors associated to a second CV event (prospective study). Please review.
  4. At the abstract, please define how study groups were defined.
  5. At the abstract, please explain how periodontitis was defined.
  6. At the abstract, please define LVEF.
  7. At the abstract, please provide general information on statistical methods used to analyze data.
  8. At Introduction, please refer to more recent studies on the association between periodontitis and MI.
  9. At Introduction, please state specific objectives.
  10. At Material and Methods, please describe the setting, locations, and relevant dates, including periods of recruitment, exposure, follow-up, and data collection.
  11. At Material and Methods, please describe inclusion/exclusion criteria for both groups.
  12. At Material and Methods, please describe how test and control groups were allocated.
  13. Please clearly describe how periodontitis was defined for the establishment of study groups. Please, also describe who performed periodontal examination, if there was previous calibration and other relevant factors for better understanding of the methods.
  14. If this is a matched-study, please give matching criteria and the number of controls per case.
  15. Please, clearly define all outcomes, exposures, predictors, potential confounders, and effect modifiers.
  16. Please, explain how quantitative variables were handled in the analyses. If applicable, describe which groupings were chosen and why.
  17. Please, report numbers of individuals at each stage of study—eg numbers potentially eligible, examined for eligibility, confirmed eligible, included in the study, completing follow-up, and analyzed.
  18. Give reasons for non-participation at each stage.
  19. Consider use of a flow diagram.
  20. For descriptive data, please give characteristics of study participants (eg demographic, clinical, social) and information on exposures and potential confounders.
  21. Please, indicate number of participants with missing data for each variable of interest.
  22. At the Results, please give unadjusted estimates and, if applicable, confounder-adjusted estimates and their precision (eg, 95% confidence interval). Make clear which confounders were adjusted for and why they were included.
  23. Please, provide information on statistical analysis at tables and text.
  24. Please, provide risk estimative for each variable.
  25. At Discussion, summarise key results with reference to study objectives.
  26. Discuss limitations of the study, taking into account sources of potential bias or imprecision. Discuss both direction and magnitude of any potential bias.
  27. Give a cautious overall interpretation of results considering objectives, limitations, multiplicity of analyses, results from similar studies, and other relevant evidence.
  28. Discuss the generalisability (external validity) of the study results.

Author Response

Answer reviewer 1

  1. Indicate the study’s design with a commonly used term in the title or the abstract. This is a retrospective case-control study followed by a prospective follow-up of the participants included. Please consult a statistics for further clarification.

The abstract has been changed according to this suggestion

  1. In the abstract, you have stated that periodontal diseases is a risk factor for cardiovascular diseases. If that is so, why are you performing this investigation?

Periodontitis and CVD share multiple risk factors; our aim with this study is to clarify if these risk factors are the same in both diseases. Therefore, we analyze 4 groups: test group with patients who suffered a MI and control group without MI, both groups were divided into patients with and without periodontitis

  1. At the abstract, please clarify the goals of this study. I could understand that this study has two major goals: (1) to investigate if periodontal disease is a risk factor for myocardial infarction in a case-control study; (2) to investigate risk factors associated to a second CV event (prospective study). Please review.

Our goals were 1) risk factors associated with both diseases: 2) risk factors associated with a second CV follow-up.

The abstract has been changed according to this suggestion.

  1. At the abstract, please define how study groups were defined.

A test group (MIG) of 144 male MI patients in the previous 48 h was compared with a control group (CG) of 138 male patients and divided according to the diagnosis of periodontitis.

  1. At the abstract, please explain how periodontitis was defined.

Periodontitis is diagnosed as stage III and IV.

  1. At the abstract, please define LVEF.

The abstract has been changed according to this suggestion.

  1. At the abstract, please provide general information on statistical methods used to analyze data.

The abstract has been changed according to this suggestion.

  1. At Introduction, please refer to more recent studies on the association between periodontitis and MI.

Recent studies have been cited.

  1. At Introduction, please state specific objectives.

Objectives have been stated.

  1. At Material and Methods, please describe the setting, locations, and relevant dates, including periods of recruitment, exposure, follow-up, and data collection.

Changes have been made according to suggestions.

  1. At Material and Methods, please describe inclusion/exclusion criteria for both groups.

As described in Materials and Methods, the inclusion criteria in MIG consisted of male patients who had a first episode of MI and in CG patients the exclusion criteria was for previous MI

  1. At Material and Methods, please describe how test and control groups were allocated.

As described, CG was matched for the same periodontal conditions as for MIG

  1. Please clearly describe how periodontitis was defined for the establishment of study groups. Please, also describe who performed periodontal examination, if there was previous calibration and other relevant factors for better understanding of the methods.

As described, periodontitis was diagnosed according to the stage III and IV definition of the 2017 World Workshop on Classification of Periodontal and Peri-Implant Diseases and Condition [22]: clinical attachment level ≥ 5 mm at 2 ≥ sites in ≥ 2 non-adjacent teeth. Severe PER was defined as 30% tooth with probing depth ≥ 4 at 2≥ interproximal sites. A calibrated explorer (PB) performed the periodontal examination for all patients.

  1. If this is a matched-study, please give matching criteria and the number of controls per case.

As described, CG were matched for the same periodontal conditions as for MIG

  1. Please, clearly define all outcomes, exposures, predictors, potential confounders, and effect modifiers.

As described in Material and Methods, clinical, diet, physical, oral biochemical, microbiological and cardiological data were analyzed.

  1. Please, explain how quantitative variables were handled in the analyses. If applicable, describe which groupings were chosen and why.

As described quantitative variables were handled with this analysis: the normal distribution of the quantitative data was verified using the Kolmogorov-Smirnov test. The statistical ANOVA test for the parametric distribution and the Mann-Whitney U and Kruskal-Wallis tests for the nonparametric distribution were used. The post hoc test utilized for the parametric distribution was t-student with the previous Levine test on homogeneity of variances and for the nonparametric distribution the Mann-Whitney U test.

  1. Please, report numbers of individuals at each stage of study—eg numbers potentially eligible, examined for eligibility, confirmed eligible, included in the study, completing follow-up, and analyzed.

As originally described, 138 patients in the CG and 144 in the MIG were explored, with a later exclusion of 26 patients in the CG and 13 patients in the MIG due to missing data.

  1. Give reasons for non-participation at each stage.

The patients were excluded due to missing data.

  1. Consider use of a flow diagram.

Perhaps a flow diagram does not clarify the methodology

  1. For descriptive data, please give characteristics of study participants (eg demographic, clinical, social) and information on exposures and potential confounders.

As described, MIG patients consisted of all male patients who had a first episode of MI in the previous 72 hours who attended the University Hospital ‘Virgen Macarena’, Sevilla.  CG patients were recruited from male patients attending the School of Dentistry of the University of Seville without a history of MI.

  1. Please, indicate number of participants with missing data for each variable of interest.

26 patients in the CG and 13 patients in the MIG due to missing data.

  1. At the Results, please give unadjusted estimates and, if applicable, confounder-adjusted estimates and their precision (eg, 95% confidence interval). Make clear which confounders were adjusted for and why they were included.

These issues are the reason to use binary logistic regression analysis. We start with all the variables in which a difference has been detected that relates to MI. The final formula removes all the confounder factors and remains only the variables that allow us to relate to MI.

  1. Please, provide information on statistical analysis at tables and text.

ANOVA, Mann-Whitney U, and Kruskal-Wallis test were added in the tables.

  1. Please, provide risk estimative for each variable.

This issue has been estimated using binary logistic regression analysis.

  1. At Discussion, summarise key results with reference to study objectives.

Our goals were to identify in patients with MI whether some general and periodontal variables can give us some new risk factors and how they can influence the prognosis of the progression of MI. Therefore, we conclude that periodontitis, exercise, diet, and smoking are risk factors related to MI. Mayor Cardiovascular Adverse Events (MACE) presented in ‘MI follow-up’ show periodontitis, weight, exercise hours, and dyslipidemia as risk factors.

  1. Discuss limitations of the study, taking into account sources of potential bias or imprecision. Discuss both direction and magnitude of any potential bias.

A new sentence has been included according to the suggestion

  1. Give a cautious overall interpretation of results considering objectives, limitations, multiplicity of analyses, results from similar studies, and other relevant evidence.

We consider that our discussion gives a cautious interpretation of the results.

  1. Discuss the generalisability (external validity) of the study results.

We consider this issue with a new sentence.

Reviewer 2 Report

The topic of the manuscript is the identification of risk factors for myocardial infarction, taking into account the general, biochemical and periodontal parameters, as well as assessment of their impact on the prognosis during convalescence in these male patients from Spain.

The title and the abstract of the article are informative. The Introduction briefly presents the issue of relationships between myocardial infarction and the presence of periodontitis, as well as the role of periopathogens in the development of atherosclerosis. The section "Material and Methods" precisely describes the chosen study design. The section "Results" is relatively clearly presented, but it should be revised from the technical side. The Discussion is interestingly written, including the recent references, however, it should be improved with the paragraph about the study limitations. The Conclusions seems to be the "take-home" messages.

Some following points must be clarified/corrected for the further processing of this article.

Merits-related comments:

  1. Please complete keywords with the proper MeSH terms necessary for indexing in the databases.
  2. Please explain why the selected control group was not matched with the study group according to age, which is a significant error and a potential source of bias, especially as they were statistically significantly younger, which may lead to better dental and periodontal status.
  3. Did the distribution of all presented biochemical parameters correspond to the normal distribution? If not, these results should be presented in the form of medians and quartiles (instead of means and standard deviation).
  4. In the rows of the table showing the percentage results, the sum does not always add up to 100% – please correct this.
  5. In the tables presenting the logistic regression results, phrases "0. 000" should be replaced with "<0. 001*". Also, I would suggest replacing exp (B) with OR (for readers with less statistical knowledge, this may not be clear).
  6. I would ask for a more detailed discussion of the results obtained in modelling the logistic regression. Please pay particular attention to the calculated odds ratio values - is it possible to state the potential risk factors if the OR for the modelled phenomenon is below zero?
  7. At the end of the Discussion, a new paragraph explaining the limitations of the study should be added.
  8. Title, abstract and conclusions may also require rewording after modification of the Results part.

Technical comments:

  1. Please check all tables with results carefully – decimal characters should be full stops (no commas). Also, all table headings should be in English (not in Spanish, e.g. "Error estándar"). All abbreviations included in the tables should be explained in the legends below.
  2. Phrases "p-value" should be used instead of "Sig." or "p". Also, statistically significant values should be marked with asterisks in Tables 6 and 7.
  3. In the text, reference numbers should be placed in square brackets [ ], and placed before the punctuation; for example [1], [1–3] or [1,3].
  4. The citation list must be corrected. References should be described as follows:
    1. Author 1, A.B.; Author 2, C.D. Title of the article. Abbreviated Journal Name YearVolume, page range.
  5. In Author Contributions, the following statements should be used "Conceptualization, X.X. and Y.Y.; Methodology, X.X.; Software, X.X.; Validation, X.X., Y.Y. and Z.Z.; Formal Analysis, X.X.; Investigation, X.X.; Resources, X.X.; Data Curation, X.X.; Writing – Original Draft Preparation, X.X.; Writing – Review & Editing, X.X.; Visualization, X.X.; Supervision, X.X.; Project Administration, X.X.; Funding Acquisition, Y.Y.".
  6. The full names of the authors should be included under the title of the manuscript.

Author Response

Merits-related comments:

  1. Please complete keywords with the proper MeSH terms necessary for indexing in the databases.

Changes have been made.

  1. Please explain why the selected control group was not matched with the study group according to age, which is a significant error and a potential source of bias, especially as they were statistically significantly younger, which may lead to better dental and periodontal status.

Periodontitis occurs mainly in the population above 40 years of age. Our data show statistical differences but clinical irrelevant, patients 61 years old are not different from 56 years of age. If our results would have shown age differences of 10 or 20 years, this could introduce an important bias. Our main goal was to identify in patients with MI whether some general and periodontal variables can give us some new risk factors, these age differences do not affect outcomes.

  1. Did the distribution of all presented biochemical parameters correspond to the normal distribution? If not, these results should be presented in the form of medians and quartiles (instead of means and standard deviation).

The statistical analysis took into consideration the normal distribution, and an appropriate test (Mann-Whitney U and Kruskal-Wallis tests) was used. If we present the variables with normal distribution with means and the variables with non-normal distribution with median and quartiles, this issue complicates and divides our table. We suggest accepting the media for the variables as an easier way to present the results.

  1. In the rows of the table showing the percentage results, the sum does not always add up to 100% – please correct this.

Sometimes the sum does not add up to 100% because the computer give us decimals that when are added are not exactly 100% but decimals around this figure.

  1. In the tables presenting the logistic regression results, phrases "0. 000" should be replaced with "<0. 001*". Also, I would suggest replacing exp (B) with OR (for readers with less statistical knowledge, this may not be clear).

Changes have been made.

  1. I would ask for a more detailed discussion of the results obtained in modelling the logistic regression. Please pay particular attention to the calculated odds ratio values - is it possible to state the potential risk factors if the OR for the modelled phenomenon is below zero?

We change the discussion to clarify the results of the logistic binary regression.

The result P is the probability that the event under investigation will occur and that the expression itself is always a number between 0 and 1, since the function e is always positive. The higher the value of Y (positive) the result of P approaches 1 and the more negative Y is the value of P approaches 0.

  1. At the end of the Discussion, a new paragraph explaining the limitations of the study should be added.

A new sentence has been added in the discussion.

  1. Title, abstract and conclusions may also require rewording after modification of the Results part.

Changes have been made.

Technical comments:

  1. Please check all tables with results carefully – decimal characters should be full stops (no commas). Also, all table headings should be in English (not in Spanish, e.g. "Error estándar"). All abbreviations included in the tables should be explained in the legends below.

Changes have been made.

  1. Phrases "p-value" should be used instead of "Sig." or "p". Also, statistically significant values should be marked with asterisks in Tables 6 and 7.

Changes have been made.

  1. In the text, reference numbers should be placed in square brackets [ ], and placed before the punctuation; for example [1], [1–3] or [1,3].

All errors have been changed.

  1. The citation list must be corrected. References should be described as follows:
    1. Author 1, A.B.; Author 2, C.D. Title of the article. Abbreviated Journal Name YearVolume, page range.
  2. Changes have been made.

  1. In Author Contributions, the following statements should be used "Conceptualization, X.X. and Y.Y.; Methodology, X.X.; Software, X.X.; Validation, X.X., Y.Y. and Z.Z.; Formal Analysis, X.X.; Investigation, X.X.; Resources, X.X.; Data Curation, X.X.; Writing – Original Draft Preparation, X.X.; Writing – Review & Editing, X.X.; Visualization, X.X.; Supervision, X.X.; Project Administration, X.X.; Funding Acquisition, Y.Y.".

Changes have been made.

  1. The full names of the authors should be included under the title of the manuscript.

Changes have been made.

Round 2

Reviewer 1 Report

Although this study is very interesting and provide novel information, especially on the association of periodontitis and recurrent cardiovascular events in a prospective follow-up of patients, major concerns still remains after the first revision:

Abstract

  1. Lines 24-26 - "A test group (MIG) of 144 male MI patients in the previous 48 h was compared with a control group (CG) of 138 male patients and divided according to the diagnosis of periodontitis." Please, consider the following suggestion for clarifying:  

    "Test group (MIG) was composed by 144 males who suffered MI in the previous 48h. Control group (CG) was composed by 138 males aged-matched with no MI. Both groups were subdivided according to the presence or not of periodontitis stages III and IV."

  2. Line 28. Please delete the sentence: 

    "Periodontitis is diagnosed as stage III and IV."

  3. Lines 30-36: "Periodontitis patients do not show differences in anthropometric variables and in previous classical risk factors. The average weekly exercise hours have a higher value in CG without periodontitis. The number of leukocytes was higher in MIG, the number of monocytes was lower in CG, the number of teeth was lower in MIG with periodontitis."

    Please clarify. No differences in anthropometric variables were found between MIG and GC periodontitis patients. Considering explaining the differences between subgroups as well.

    What do you mean by "in previous classical risk factors"? Please describe.

    Increased countings of leukocytes were observed in MIG (both with and without periodontitis?); GC showed lower countings of monocytes. Please note that Table 1 shows higher countings of monocytes in GC. Please review.

    Please describe data according to the category: anthropometric values, biochemical analysis, microbiological analysis, periodontal parameters, etc. Do not mix information in the same sentence/paragraph.

  4. Lines 39-41: 

    "Follow-up LVEF values are preserved in patients with non-periodontitis without periodontitis. Our data suggest that periodontitis can be considered as a risk factor for MI and MACE in the studied population. Periodontitis should be considered an important risk factor in MI follow-up.

    Introduction
  5. Lines 51-52: 

    "One of the most prevalent inflammatory diseases is periodontal disease, periodontitis (PER) is considered the sixth most common disease in the world, a group of bacteria has been associated with alveolar bone resorption and loss of all teeth [3]." Substitute for:

    "Periodontitis (PER) is an infectious inflammatory disease caused by an interaction between host responses and microbial dysbiosis, resulting in alveolar bone loss and may leading to tooth loss, if left untreated (3)."

  6. Lines 53-56: "Observational studies support an association with cardiovascular disease between periodontitis and cardiovascular diseases [4-6]. However, previous other studies have shown controversial results [7,8], which could impact public health policies if new risk factors for MI are identified.

    some of them demonstrate a relationship 54 independent of known confounders [7], but others have not [8], and this controversy is very  important given the potential impact on public health if a new risk factor could be identified." 
  7. Lines 57-60: 

    "PER is associated with multiple bacterial species, mainly Bacteroidetes and, more specifically, with the genera  of Porphyromonas, Prevotella, Tannerella, and Capnocytophaga genera. These bacteria are involved in the development of atherosclerosis in experimental animals [9] and are related associated to symptomatic atherosclerosis [10] and 59 may correlate with disease its markers of atherosclerosis [11] in humans."

  8. Lines 60-61. The following sentence seems to be incomplete. Please review.

    "We must emphasize that the oral cavity is part of the digestive tract and the high level of inter-individual contamination through mouth-to-mouth contact [12]."

  9. Lines 62-66: "In recent the past few decades, incredible effort has been made to improve our knowledge on the risk factors associated with myocardial infarction. However, the pathogenic mechanisms involved in atherosclerosis development remains unclear. many open questions to explain all the details of the mechanisms that avoid the development of atherosclerosis. One of the most important ways to prevent MI and one of the most effective strategies to do this is to control cardiovascular risk factors [13]. These include nutrition, physical activity, smoking, hypertension, and diabetes."
  10. Lines 76-78: "Our goals was were (1) to identify risk factors associated with myocardial infarction in males with and without periodontitis and (2) to investigate which risk factors in patients with MI whether some  general and periodontal variables can give us some new risk factors and how they can  influence the prognosis of the progression of MI."

Material and Methods

  1. Lines 82-84: 

    "The test group (MIG) consisted of male patients who had a first episode of MI in the pre-82 vious 72 hours who attended the University Hospital ‘Virgen Macarena’, Sevilla, from 83 February 2017 to February 2019."

    This is not in accordance with the information provided in the abstract. Please note that, at the abstract, participants included in test group had to present with myocardial infarction in the previous 48 hours. Please correct.

  2. Lines 88-89.

    "Both groups were divided into two other groups 88 according to the presence or absence of PER."

    Please, describe here how periodontitis was defined.
  3. Line 90: "CGs were matched for the same periodontal conditions as for MIG."  This introduces a bias, since it is not able to verify differences in the prevalence, severity or extension of periodontal diseases in a random population. Besides that, table 1 shows lesser number of teeth in MIG with periodontitis, and table 6 indicates that periodontitis increases the risk of MI based on the prevalence of CAL ≥ 5 mm, which suggests that periodontal conditions were not the same in the different groups. So far, I could understand that patients were included in MIG if they had miocardyal infarcton in the previous 48 or 72h, being this the primary outcome. In addition to this, patients were subdivided in two subgroups, according to the presence or absence of periodontitis. I suggest deleting the sentence describing that groups were matched according to periodontal conditions. I do understand that what you mean is that patients were subdivided in both test and control groups according to the presence or absence of periodontal diseases, but the sentence is unclear in the text. Please revise.

  4. Line 115. 

    Ainamo and Bay recorded BOP 10 seconds (not 30 seconds) after introducing the probe into the pocket. Please correct.

  5. Lines 131-133. 

    "After the supragingival plaque removal and isolation of the area, the paper points were subgingival inserted into gingival sulcus for 30 seconds. , then Samples were pooled, stored in an Eppendorfs tube, and immediately processed immediately."

  6. Line 747: At baseline, t he MIG base visit, the following cardiovascular risk factors were recorded in MIG:

RESULTS

  1. Lines 173-176: 

    "Initially, 138 patients in the CG and 144 in the MIG were explored, with a later exclusion 173 of 26 patients in the CG and 13 patients in the MIG due to missing data. The final sample 174 size for CG was 112 patients (66 with PER and 46 without periodontitis (NOPER) and for 175 MIG 131 patients (85 with PER and 46 NOPER). During the MIG follow-up, 1 patient 176 died and 4 patients were lost."

    Please provide a flow diagram with this information.
  2. Table 1: 

    (a) Please indicate the statistical test used to analyze data by using signals and describing them at the legends.

    (b) Please note that CG is significantly older than MIG. The prevalence, severity and extension of periodontitis increases with age. Considering that, it can be expected that one or more of these parameters may be increased in CG when compared with MIG.

    (c) Monocytes: At the abstract, there is an information that  the number of monocytes are lower is GC, but the table demonstrates that the average number of monocytes are increased in GC compared with MIG. Please correct/clarify.

    (d) Smoking and MDA: please indicate statistical teste used.
  3. Please, it is essential that you provide a Table describing periodontal parameters in each study group, with statistical analysis.
  4. Table 5. At second column (Category), line corresponding to LVEF at admission, the last line would be severely reduced?
  5. Table 6. What do you mean by "B" in the second column?

Discussion

  1. Line 245. 

    "Our results show that periodontitis is related to MI as demonstrated in the literature [24, 25, 245 26]." 

    These data is not demonstrated, since no table describing periodontal parameters found in MIG and CG was provided. Besides that, OR values for periodontitis parameters are lower than hours of exercise, mediterranean diet and smoking. Please provide those information, comparing groups by ANOVA post hoc Tukey for linear measurments and Kruskal Wallis post hoc Dunn for non-linear variables.

2. Lines 247-252: 

Please describe this at Material and Methods - Statistical analysis section.

3. Line 253: 

"between periodontitis and no periodontitis patients, including:"

4. Lines 255-256: please clarify.

5. Line 271: 

"the chronic inflammation condition due to more severe periodontal disease". Data not demonstrated.

6. Line 276: 

between the number of teeth - Data not demonstrated at Tables 6 or 7

7. Lines 281-282: 

Please note that all participants were overweight (average values) in spite of exercising hours/week. Besides that, although no differences were found between groups, HbA1c levels in GC are compatible with normoglicemic patients while these parameters in MIG are compatible with prediabetes or insulin resistance. Also, hypertension was investigated and this could also, in the presence of increased abdominal circumference, diagnose metabollic syndrome. These findings should be better explored by the authors.

8. Line 287: 

Please note that your study did not evaluate Eikenella corrodens. Since that, "on the contrary" should not be used here.

Author Response

Abstract

  1. Lines 24-26 - "A test group (MIG) of 144 male MI patients in the previous 48 h was compared with a control group (CG) of 138 male patients and divided according to the diagnosis of periodontitis." Please, consider the following suggestion for clarifying:  

"Test group (MIG) was composed by 144 males who suffered MI in the previous 48h. Control group (CG) was composed by 138 males aged-matched with no MI. Both groups were subdivided according to the presence or not of periodontitis stages III and IV."

New sentence has been included

  1. Line 28. Please delete the sentence: 

"Periodontitis is diagnosed as stage III and IV."

Sentence has been deleted

  1. Lines 30-36: "Periodontitis patients do not show differences in anthropometric variables and in previous classical risk factors. The average weekly exercise hours have a higher value in CG without periodontitis. The number of leukocytes was higher in MIG, the number of monocytes was lower in CG, the number of teeth was lower in MIG with periodontitis."

Please clarify. No differences in anthropometric variables were found between MIG and GC periodontitis patients. Considering explaining the differences between subgroups as well. Change has been made according to the suggestions

What do you mean by "in previous classical risk factors"? Please describe. There are hypertension, diabetes, dyslipidemia and smoking

Increased countings of leukocytes were observed in MIG (both with and without periodontitis?); GC showed lower countings of monocytes. Please note that Table 1 shows higher countings of monocytes in GC. Please review. Change has been made

Please describe data according to the category: anthropometric values, biochemical analysis, microbiological analysis, periodontal parameters, etc. Do not mix information in the same sentence/paragraph. Sentences have been reviewed

  1. Lines 39-41: 

"Follow-up LVEF values are preserved in patients with non-periodontitis without periodontitis. Our data suggest that periodontitis can be considered as a risk factor for MI and MACE in the studied population. Periodontitis should be considered an important risk factor in MI follow-up.

Changes have made according to the suggestions

Introduction

  1. Lines 51-52: 

"One of the most prevalent inflammatory diseases is periodontal disease, periodontitis (PER) is considered the sixth most common disease in the world, a group of bacteria has been associated with alveolar bone resorption and loss of all teeth [3]." Substitute for:

"Periodontitis (PER) is an infectious inflammatory disease caused by an interaction between host responses and microbial dysbiosis, resulting in alveolar bone loss and may leading to tooth loss, if left untreated (3)." New sentence has been included

  1. Lines 53-56: "Observational studies support an association with cardiovascular disease between periodontitis and cardiovascular diseases [4-6]. However, previous other studies have shown controversial results [7,8], which could impact public health policies if new risk factors for MI are identified.

some of them demonstrate a relationship 54 independent of known confounders [7], but others have not [8], and this controversy is very  important given the potential impact on public health if a new risk factor could be identified."

New sentence has been included

  1. Lines 57-60: 

"PER is associated with multiple bacterial species, mainly Bacteroidetes and, more specifically, with the genera  of PorphyromonasPrevotellaTannerella, and Capnocytophaga genera. These bacteria are involved in the development of atherosclerosis in experimental animals [9] and are related associated to symptomatic atherosclerosis [10] and 59 may correlate with disease its markers of atherosclerosis [11] in humans."

Changes have made according to the suggestions

  1. Lines 60-61. The following sentence seems to be incomplete. Please review.

"We must emphasize that the oral cavity is part of the digestive tract and the high level of inter-individual contamination through mouth-to-mouth contact [12]."

Sentence has been changed

  1. Lines 62-66: "In recent the past few decades, incredible effort has been made to improve our knowledge on the risk factors associated with myocardial infarction. However, the pathogenic mechanisms involved in atherosclerosis development remains unclear. many open questions to explain all the details of the mechanisms that avoid the development of atherosclerosis. One of the most important ways to prevent MI and one of the most effective strategies to do this is to control cardiovascular risk factors [13]. These include nutrition, physical activity, smoking, hypertension, and diabetes."

Changes have been made

  1. Lines 76-78: "Our goals was were (1) to identify risk factors associated with myocardial infarction in males with and without periodontitis and (2) to investigate which risk factors in patients with MI whether some  general and periodontal variables can give us some new risk factors and how they can  influence the prognosis of the progression of MI."

Changes have been made

Material and Methods

  1. Lines 82-84: 

"The test group (MIG) consisted of male patients who had a first episode of MI in the pre-82 vious 72 hours who attended the University Hospital ‘Virgen Macarena’, Sevilla, from 83 February 2017 to February 2019."

This is not in accordance with the information provided in the abstract. Please note that, at the abstract, participants included in test group had to present with myocardial infarction in the previous 48 hours. Please correct. Change has been made

  1. Lines 88-89.

"Both groups were divided into two other groups 88 according to the presence or absence of PER."

Please, describe here how periodontitis was defined.

Periodontitis is defined later in the Oral examination paragraph, but according to the suggestion it is included in the Study Participants paragraph

  1. Line 90: "CGs were matched for the same periodontal conditions as for MIG."  This introduces a bias, since it is not able to verify differences in the prevalence, severity or extension of periodontal diseases in a random population. Besides that, table 1 shows lesser number of teeth in MIG with periodontitis, and table 6 indicates that periodontitis increases the risk of MI based on the prevalence of CAL ≥ 5 mm, which suggests that periodontal conditions were not the same in the different groups. So far, I could understand that patients were included in MIG if they had miocardyal infarcton in the previous 48 or 72h, being this the primary outcome. In addition to this, patients were subdivided in two subgroups, according to the presence or absence of periodontitis. I suggest deleting the sentence describing that groups were matched according to periodontal conditions. I do understand that what you mean is that patients were subdivided in both test and control groups according to the presence or absence of periodontal diseases, but the sentence is unclear in the text. Please revise.

I agree it could be unclear, so the sentence has been deleted

  1. Line 115. 

Ainamo and Bay recorded BOP 10 seconds (not 30 seconds) after introducing the probe into the pocket. Please correct.

It has been corrected

  1. Lines 131-133. 

"After the supragingival plaque removal and isolation of the area, the paper points were subgingival inserted into gingival sulcus for 30 seconds. , then Samples were pooled, stored in an Eppendorfs tube, and immediately processed immediately."

Changes have been included

  1. Line 747: At baseline, t he MIG base visit, the following cardiovascular risk factors were recorded in MIG:

Changes have been made

RESULTS

  1. Lines 173-176: 

"Initially, 138 patients in the CG and 144 in the MIG were explored, with a later exclusion 173 of 26 patients in the CG and 13 patients in the MIG due to missing data. The final sample 174 size for CG was 112 patients (66 with PER and 46 without periodontitis (NOPER) and for 175 MIG 131 patients (85 with PER and 46 NOPER). During the MIG follow-up, 1 patient 176 died and 4 patients were lost."

Please provide a flow diagram with this information. A flow diagram has been provided

  1. Table 1: 

(a) Please indicate the statistical test used to analyze data by using signals and describing them at the legends. It has been added

(b) Please note that CG is significantly older than MIG. The prevalence, severity and extension of periodontitis increases with age. Considering that, it can be expected that one or more of these parameters may be increased in CG when compared with MIG.

Periodontitis occurs mainly in the population above 40 years of age. Our data show statistical differences but clinical irrelevant, patients 61 years old are not different from 56 years of age. If our results would have shown age differences of 10 or 20 years, this could introduce an important bias. Our main goal was to identify in patients with MI whether some general and periodontal variables can give us some new risk factors, these age differences do not affect the outcomes.

(c) Monocytes: At the abstract, there is an information that  the number of monocytes are lower is GC, but the table demonstrates that the average number of monocytes are increased in GC compared with MIG. Please correct/clarify. It has been corrected

(d) Smoking and MDA: please indicate statistical teste used. It has been added

  1. Please, it is essential that you provide a Table describing periodontal parameters in each study group, with statistical analysis.

Table 2 has been added

  1. Table 5. At second column (Category), line corresponding to LVEF at admission, the last line would be severely reduced?

Changes have been made

  1. Table 6. What do you mean by "B" in the second column?

The values of the exponent of the formula are detailed in column B. Bi are the values for each variable of the model and Xi is the value of the independent variable to which Bi corresponds and the independent value (without Xi) is the value of column B for the constant.

Discussion

  1. Line 245. 

"Our results show that periodontitis is related to MI as demonstrated in the literature [24, 25, 245 26]." 

These data is not demonstrated, since no table describing periodontal parameters found in MIG and CG was provided. Besides that, OR values for periodontitis parameters are lower than hours of exercise, mediterranean diet and smoking. Please provide those information, comparing groups by ANOVA post hoc Tukey for linear measurments and Kruskal Wallis post hoc Dunn for non-linear variables.

The prognosis on the probability of infarction will be automatically completed with an RLB model in which severe periodontitis was assigned the code 1 to not suffer from periodontitis and 0 to suffer from it, so the value of B has the same meaning as the number of sites with CAL≥5 mm, Average weekly exercise hours and Percentage MDA. Change to no severe periodontitis have been made. Therefore, no suffer periodontitis decreases the risk of infarction and smoking increases de risk of infarction.

  1. Lines 247-252: 

Please describe this at Material and Methods - Statistical analysis section.

Change has been made

  1. Line 253: 

"between periodontitis and no periodontitis patients, including:"

Change has been made

  1. Lines 255-256: please clarify.

The sentence has been modified

  1. Line 271: 

"the chronic inflammation condition due to more severe periodontal disease". Data not demonstrated.

Sentence has been changed according to the suggestion

  1. Line 276: 

between the number of teeth - Data not demonstrated at Tables 6 or 7

It has been deleted

  1. Lines 281-282: 

Please note that all participants were overweight (average values) in spite of exercising hours/week. Besides that, although no differences were found between groups, HbA1c levels in GC are compatible with normoglicemic patients while these parameters in MIG are compatible with prediabetes or insulin resistance. Also, hypertension was investigated and this could also, in the presence of increased abdominal circumference, diagnose metabollic syndrome. These findings should be better explored by the authors.

This is the main finding of our data. Metabolic syndrome is defined with the parameters: waist circumference, glycaemia ≥100mgr/dl, arterial tension ≥130/85, triglycerides≥ 150mgr/dl and HDL cholesterol ≤50mgr/dl. According to the definition of the International Diabetes Federation. 2021 (IDF Diabetes Atlas, 10th edn. 10th ed. Brussels, Belgium) prediabetes is defined with HbA1c 5,7-6,4%. All the groups show these data, considered as risk factors for MI, as pathological, and there are not differences between groups. So, patients with similar pathological systemic conditions some of them produced a MI and other not. We provide data in table 7 and 8 of other risk factors that must be taken in account to prevent MI and adverse follow-up complications.

We change some sentences to highlight these aspects

  1. Line 287: 

Please note that your study did not evaluate Eikenella corrodens. Since that, "on the contrary" should not be used here.

We change the sentence according to the suggestion

Reviewer 2 Report

The Authors have only referred to some of the comments and the manuscript needs further revisions.

I would suggest Authors consult the statistician as the answers to the Results part are quite confused. Both mentioned tests (Mann-Whithey and Kruskal-Wallis) are non-parametric tests, i. e. for variables that do not correspond to the normal distribution. Therefore, it was suggested to present the results in the form of medians with quartiles instead of means with standard deviations. Also, approximate values should be adjusted so that the overall results are 100%. Still, not all incorrect commas in the result values have been replaced by full stops (e. g. Table 7).

Again, please pay particular attention to the calculated odds ratio values - is it possible to state the potential risk factors if the OR for the modelled phenomenon is below 1 (between 0 and 1)? Above 1 the odds increase and below 1 decrease…

Moreover, I would suggest expanding the paragraph on the limitations of the study.

The article also requires considerable editorial editing (e. g. references, tables).

Author Response

I would suggest Authors consult the statistician as the answers to the Results part are quite confused. Both mentioned tests (Mann-Whithey and Kruskal-Wallis) are non-parametric tests, i. e. for variables that do not correspond to the normal distribution. Therefore, it was suggested to present the results in the form of medians with quartiles instead of means with standard deviations.

Medians and quartile have been included in table 1 and 3

 Also, approximate values should be adjusted so that the overall results are 100%.

Results have adjusted to 100%

Still, not all incorrect commas in the result values have been replaced by full stops (e. g. Table 7).

Commas have been replaced by full stops

Again, please pay particular attention to the calculated odds ratio values - is it possible to state the potential risk factors if the OR for the modelled phenomenon is below 1 (between 0 and 1)? Above 1 the odds increase and below 1 decrease…

The prognosis on the probability of infarction will be automatically completed with an RLB model in which severe periodontitis was assigned the code 1 to not suffer from periodontitis and 0 to suffer from it, so the value of B has the same meaning as the number of sites with CAL≥5 mm, Average weekly exercise hours and Percentage MDA. Change to no severe periodontitis have been made. Therefore, no suffer periodontitis decreases the risk of infarction and smoking increases de risk of infarction.

Moreover, I would suggest expanding the paragraph on the limitations of the study. We add new sentences at the end of discussion

The article also requires considerable editorial editing (e. g. references, tables).

The paper has been thoroughly reviewed